# Seroprevalence and incidence of hepatitis A in Southeast Asia: A systematic review

**Gustavo Hernandez-Suarez**[1]*, **Debasish Saha**[1], **Kris Lodroño**[2], **Phatu Boonmahittisut**[3], **Stephanus Taniwijaya**[4], **Ashwini Saha**[5], **Selim Badur**[6], **Yong Poovorawan**[7]

**1** GSK, Medical & Clinical Emerging Markets, Wavre, Belgium, **2** GSK, Medical Affairs Philippines, Manila, The Philippines, **3** GSK, Thailand Vaccines, Bangkok, Thailand, **4** GSK, Medical & Clinical Emerging Markets, Jakarta, Indonesia, **5** GSK, Vaccines Medical, Kuala Lumpur, Malaysia, **6** GSK, Medical & Clinical Emerging Markets, Istanbul, Turkey, **7** Center of Excellence in Clinical Virology, Faculty of Medicine, Chulalongkorn University, Bangkok, Thailand

\* gustavo.x.hernandez@gsk.com

## Abstract

### Background

A previous review on hepatitis A virus (HAV) seroprevalence in 2005 categorized Southeast Asia as a low HAV endemicity region. In 2010, the World Health Organization modified this from low to low/medium endemicity, pointing out that these estimates were based on limited evidence. Since then, there has been no attempt to review HAV epidemiology from this region. We conducted a systematic review of literature to collect information on HAV incidence and seroprevalence in select countries in the Southeast Asian region, specifically, The Association of Southeast Asian Nations over the last 20 years.

### Methodology

This systematic review was conducted according to the Preferred Reporting Items for Systematic Reviews and Meta-analyses guidelines. From the relevant articles, we extracted data and conducted a risk of bias assessment of individual studies.

### Results

The search yielded 22 and 13 publications on HAV seroprevalence and incidence, respectively. Overall, our findings point to a very low HAV endemicity profile in Thailand and Singapore and evidence of a shift towards low HAV endemicity in Indonesia, Lao People's Democratic Republic, Malaysia, the Philippines, and Vietnam. Only Singapore, Thailand, Malaysia, and the Philippines have existing HAV disease surveillance and reported incidence rates below 1 per 100,000. Several outbreaks with varying magnitude documented in the region provide insights into the evolving epidemiology of HAV in the region. Risk of bias assessment of studies revealed that the individual studies were of low to medium risk.

### Conclusions/Significance

The available HAV endemicity profiles in Southeast Asian countries, aside from Thailand, are limited and outdated, but suggest an endemicity shift in the region that is not fully

**Data Availability Statement:** All relevant data are within the manuscript and its Supporting Information files.

**Funding:** GHS, DS, KL, PB, ST, AS and SB received funding in the form of salary from the GSK group of companies. GlaxoSmithKline Biologicals SA employees were involved in the study design, data collection and analysis, decision to publish, and preparation of the manuscript. The specific roles of these authors are articulated in the 'author contributions' section.

**Competing interests:** GHS, DS, KL, PB, ST, AS and SB are employees of the GSK group of companies. GHS and DS hold shares in the GSK group of companies. YP was supported by The Center of Excellence in Clinical Virology, Chulalongkorn University. Authors declare no other financial and non-financial relationships and activities. GlaxoSmithKline Biologicals SA funded this review and all costs associated with its development and publication. All authors had full access to all of the data in this study and take complete responsibility for the integrity of the data and accuracy of the data analysis. We declare that HavrixTM, an inactivated hepatitis A vaccine, is a trademark of the GSK group of companies that is marketed in several countries in Southeast Asia under the scope of this review. There are no additional patents, products in development or marketed products to declare. This does not alter our adherence to PLOS ONE policies on sharing data and materials.

documented yet. These findings highlight the need to update information on HAV epidemiology through strengthening of disease surveillance mechanisms to confirm the shift in HAV endemicity in the region.

## Introduction

Hepatitis A disease is caused by the hepatitis A virus (HAV) and is transmitted via the fecal-oral route either through ingestion of contaminated food or water, or through close contact with an infectious person [1, 2]. Symptoms are initially non-specific (nausea, vomiting, fever, malaise and abdominal pain) and followed by bilirubinuria, pale stools, jaundice (usually 2–4 weeks) and pruritus (unusual case) that can last up to 6 months [3]. The severity and clinical outcome of hepatitis A are highly correlated with age at infection, with HAV infection being usually asymptomatic in children (~70% of cases) but commonly symptomatic in adults (>70% of cases) [3]. The presence of IgG antibodies confirms either vaccination or the initiation of the convalescent phase of infection, providing lifelong immunity to the individual [3]. Endemicity level (i.e. circulation) of HAV within a given region or population is readily estimated through serological surveys (measurement of HAV antibodies in the blood) and are reliable estimates of the burden of disease in a population.

HAV endemicity is typically high in low- and middle-income countries (where individuals are exposed to virus infection in their childhood years) in comparison to the situation in high-income countries which typically have very low endemicity (i.e. most of the population is naïve to HAV infection during their life) [4]. In recent decades, low- and middle-income countries have seen rising incomes and rapid urbanization and nearly the entire population (especially in urban areas) now have access to clean water. Consequently, these countries have reported a transition from high to low endemicity levels of HAV due to a progressive decrease of exposure to HAV during their childhood [4]. Paradoxically, this improvement place countries at higher risk for HAV [5–7]. Lastly, international trade, travel, and migration may also add to the high risk of outbreaks in countries reporting low or intermediate HAV endemicity [8].

A review investigating the seroprevalence of HAV in children and adolescents in the Southeast Asian region was published in 1998 [9]. Since then, no published review investigating exposure to HAV has been identified for the region. In 2005, a systematic review of the global seroprevalence of HAV ranked the Southeast Asian region as one with low endemicity, mainly driven by a large number of publications coming from Thailand [10]. In 2010, the World Health Organization (WHO) updated the aforementioned ranking and modified the HAV endemicity status for this region to low/medium endemicity [5], but this estimation was based on limited evidence with an average of less than one publication per country. A recent systematic review reported HAV seroepidemiology in the Asia-Pacific region, including all the East and South Asian countries [11]. However, results from only 3 out of 11 countries in the Southeast Asia Region (Thailand, Indonesia, and Singapore) were reported.

In this systematic review we evaluate the existing literature regarding HAV epidemiology by focusing on disease incidence and trends of exposure to HAV in the Southeast Asia region over the last 20 years. Specifically, the 11 members of the Association of Southeast Asian Nations (ASEAN) are included in the review (Brunei, Cambodia, Indonesia, Lao People's Democratic Republic [PDR], Malaysia, Myanmar, the Philippines, Singapore, Thailand, Timor-Leste and Vietnam).

## Methods

### Search sources and strategy

The search was conducted in 5 electronic databases. We searched MEDLINE (via PubMed), Embase, Google Scholar and 2 regional databases: the Health Research and Development Information Network (HERDIN) from The Philippines [12] and MyJurnal from Malaysia [13]. We also consulted grey literature sources such as the official websites from the Ministry of Health (MoH) or ministry of public health (MOPH) and the National Health Agency of each country (including the official national report of waterborne diseases if available). To intensify the grey literature search we sent an information request letter to universities and research centers, known for their research activity in infectious diseases, requesting unpublished reports on HAV epidemiology. Lastly, we conducted a snowball search utilizing the bibliographies of the identified publications to retrieve any further relevant studies.

The full electronic search strategy is described in the **S1 Text**. Searches were limited to publications from January 01, 1999 to February 15, 2021 covering the last two decades of publications and conducted in both English and the local language of the country of interest.

### Screening and selection

The identified publications were screened using the inclusion and exclusion criteria provided in **S1 Table**. Original research from non-interventional studies was included if they reported the incidence of hepatitis A (usually defined as patients registered in clinical records or surveillance systems with the code B15 according to the International Classification of Diseases, Tenth Revision [ICD-10]) or the age-specific seroprevalence of HAV (defined as previous exposure to HAV confirmed by laboratory detection of HAV IgM/IgG in blood samples). Case reports and other publications such as commentaries, editorials and letters were excluded from this review. Both screening and selection of publications was performed by one author (GHS).

### Risk of bias

The quality of individual studies included in this review from peer-reviewed sources was assessed using the tool for critical appraisal of prevalence studies in health research literature described by Hoy *et al.* [14]. The risk of bias assessment for each paper was performed independently by two reviewers (first reviewers KL, PB, ST, AS; second reviewers: DS, SB). Any disagreement was resolved by a third reviewer (GHS).

Risk of bias was not assessed for reports or publications from official MoH websites and outbreak specific studies. This was done because insufficient data and potentially biased disease reporting levels are known to exist in such reports as seen in other countries (such as United States) [15]. Thus, conducting a risk of bias assessment for these publications was considered out of the scope for this review.

### Data extraction, statistical analysis and reporting

The information extracted from selected studies included study characteristics (year of publication, study design, main objective of the study and sample size), age group of the study population and case definition (laboratory confirmation methods and clinical definition of hepatitis A in the case of an outbreak study). Incidence cases were reported in numbers when available. Crude incidence rates and age specific seroprevalence were extracted and reported as such. If only incident cases were reported, the crude incidence rates were estimated for comparative purposes using the official population estimate from the corresponding National Statistics Bureau of each country.

Annual incidence rates and age specific seroprevalence (by country or subregion) were plotted to describe the observed trend. HAV country endemicity was classified according to WHO criteria as follows: high (for countries reporting a seroprevalence above 90% by the time individuals were 10 years of age); intermediate (seroprevalence $\geq$50% by 15 years of age but <90% by 10 years of age); low (seroprevalence $\geq$50% by 30 years of age, but <50% by 15 of age); and very low (<50% by 30 years of age) [16]. A descriptive analysis of the extracted data and risk of bias assessment were performed to summarize the main outcomes. As heterogeneity of results was expected in the study populations due to differences in socioeconomic development and surveillance systems, no meta-analysis was planned for this review.

## Results

### Search results

Databases and snowball search yielded 448 publications. After removing 201 duplicates, and exclusions, a total of 35 publications were included in the final review (**Fig 1**).

### Study characteristics

Among the 35 included publications, the largest number were for Thailand (n = 20) [23–38, 44–46, 51] followed by Malaysia (n = 5) [20, 41–43, 47]. While most countries contributed

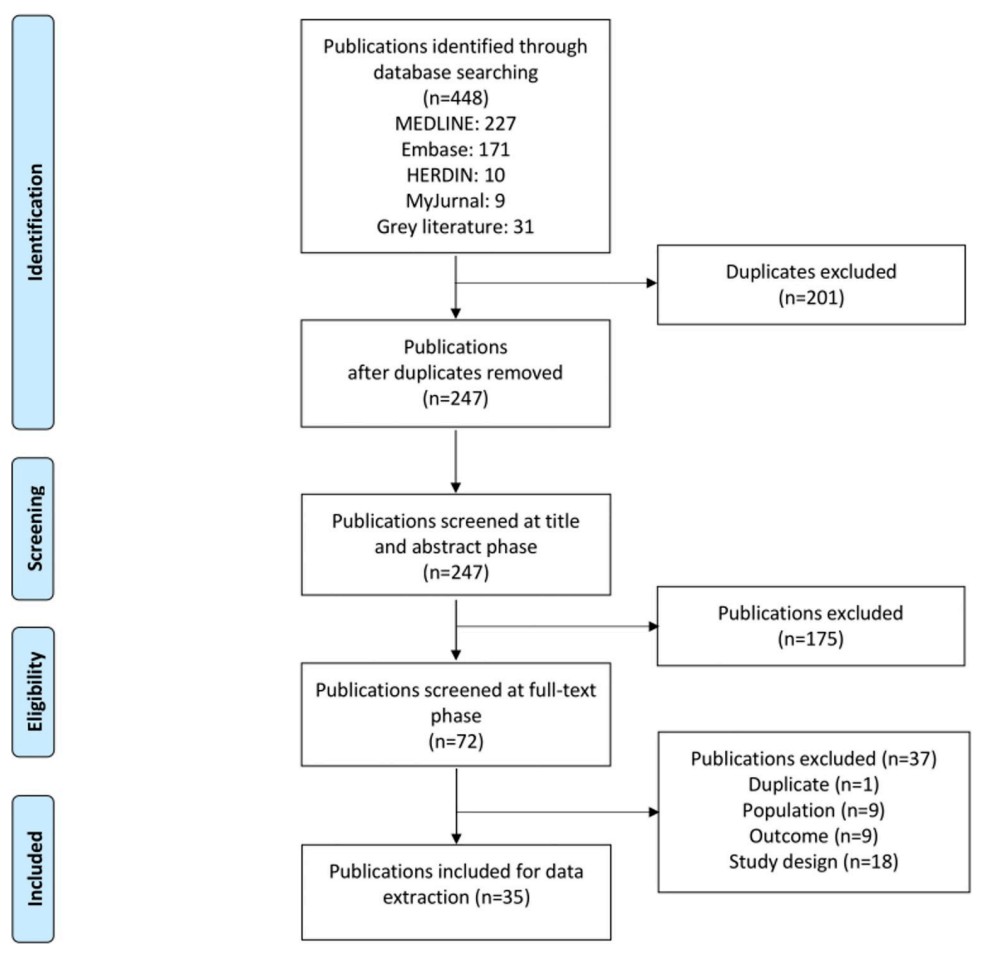

**Fig 1. PRISMA flow diagram.**

with a single study to this review, two studies provided data for immigrants coming from Vietnam, Philippines, Cambodia, Lao PDR and Myanmar [18, 21], no publications were found from Timor-Leste and Brunei. Twenty-two cross-sectional studies reported seroprevalence data while 13 publications reported only incidence data. Eight publications provided data from outbreak investigations and 5 publications were from official surveillance reports. Considering the study settings we found most reports community based (n = 10), followed by hospital based (n = 7) laboratory population (**Table 1**).

## HAV seroprevalence

Among the 22 publications 15 reported results from Thailand and were mostly conducted before 2010 (**Table 1**). Seroprevalence surveys had sample sizes ranging between 100 and 4,260 individuals.

**Thailand.** At least 3 surveys conducted in Thailand in the study period [23, 29, 37] spanning the population of 12 different provinces may be representative of the national level (**Fig 2A**). These surveys independently showed a sharp increase in HAV exposure with age: from 20% or less in population <2 years of age to more than 90% in those ≥60 years of age. When comparing across surveys over time, we observed that exposure to HAV in individuals 15–20 years of age halved in the study period: from nearly 20% (1999) to less than 10% in 2014. It was also documented that the 50% exposure to HAV threshold was reached at 42 years of age in 2014 compared to 36 years of age in 2004, thus confirming the HAV epidemiological transition of Thailand into a very low endemicity country. During the same period, data from Bangkok focusing on chronic liver disease patients [33], intravenous drug users [27] and men having sex with men [26] showed a similar age pattern (**Fig 2B**).

The latest publication from Thailand analyzed exposure to HAV in older adults (≥60 years of age) and found it to be close to 100% [24]. In Thailand, seroprevalence surveys also depicted regional variability. In the exposure to HAV threshold of 50% was reached by the age of 15 years [38], while in the province of Chiang Rai, this threshold was reached by the age 50 years [29]. These results were in high contrast with the results from other regions which were closer to the nationwide average (**Fig 2C**). Results from 5 other studies from Thailand provided estimates of HAV exposure among medical students and medical cadets [28, 36, 52] and at specific age groups [30, 35].

**Other countries.** Data from other countries included in the systematic review suggest high heterogeneity on the endemicity level of HAV across the region (**Fig 3**).

In Singapore, a population-based survey conducted in 1993 showed a HAV exposure level of 50% by 35 years of age, suggesting a very low endemicity profile [22]. In 1993 a survey ran in Yogyakarta, Indonesia [19] also showed an increase in exposure to HAV with increasing age, reaching the threshold of 50% at around 25 years of age, indicating low HAV endemicity. On the contrary, a survey from Vietnam [39], conducted in 1994, showed an almost complete seroconversion by 10 years of age. Twenty years later, a survey conducted between 2011 and 2017 among immigrant women in Korea [21] from countries including Cambodia (n = 58), the Philippines (n = 85) and Vietnam (n = 422), suggested an HAV endemicity age shift pattern in Vietnam (60% by 20–29 years of age and 80% by 30–40 years of age) and the Philippines (78% by 20–29 years of age and 72% by 30–40 years of age), but not in Cambodia (100% HAV seropositivity by 20–29 years of age).

A recent study from Lao PDR suggests a shift taking place in that country towards a low HAV endemicity profile, suggesting exposure to HAV of 50% among individuals around 20 years of age [17].

**Table 1. Study characteristics of included studies (n = 35).**

| Author | Country | Publication year | Study year | Study design | Study population | Study setting | Age group (years) | Outcome | Clinical measurement | Platform manufacturer | Total number of samples tested | Seroprevalence |
|---|---|---|---|---|---|---|---|---|---|---|---|---|
| Khounvisith et al. [17] | Lao PDR | 2020 | 2013–2018 | Cross-Sectional | Population based/ Multiple cities | Urban/semi-urban (Peak district) or rural (Phasay district) | all ages | Seroprevalence | Anti-HAV IgG | Diasorin | 1,195 | 51.05% |
| Poovorawan et al. [18] | Cambodia, Lao PDR, Myanmar | 2009 | 2008 | Cross-Sectional | Legal Immigrant workers in Thailand/ Single city | Hospital based, Bangkok | 16–60 | Seroprevalence | Anti-HAV IgG antibody | Murex Biotech | 1,183 | 95.01% |
| Juffrie et al. [19] | Indonesia | 2000 | 1995–1996 | Cross-Sectional | Population based/ Single city | Regional (Gondokusuman) community based | 4–32 | Seroprevalence | Anti-HAV IgG antibody | Abbott | 1,103 | 28.65% |
| Ahmad et al. [20] | Malaysia | 2011 | 2009 | Cross-Sectional | CLD patients/ Single City | Hospital based (Gastroenterology Clinic of Universiti Sains Malaysia, Kelantan) | ≥21 | Seroprevalence | Anti-HAV IgG antibody | Abbott | 119 | 88.24% |
| Kwon et al. [21] | Philippines, Vietnam | 2018 | 2011–2017 | Cross-Sectional | Immigrant females to Korea/ Nationwide | Regional screening programs from different provinces in Korea | 20–40 | Seroprevalence | Anti-HAV IgG antibody | bioMérieux | 575 | 73.91% |
| Lee et al. [22] | Singapore | 2011 | 1993 | Cross-Sectional | n.a. | n.a. | all ages | Seroprevalence | Anti-HAV IgG antibody | Abbott | 930 | 25.91% |
| Sa-nguanmoo et al. [23] | Thailand | 2016 | 2014 | Cross-Sectional | Pediatric health check-up or outpatient clinic at hospitals/ Nationwide | Hospital based/ Seven provinces representing four regions of Thailand | >71 | Seroprevalence | Anti-HAV IgG antibody | Abbott | 4,260 | 34.53% |
| Posuwan et al. [24] | Thailand | 2019 | 2017 | Cross-Sectional | Single city (no further details provided) | Participants were from the Chum Phae district of Khon Kaen province in northeastern Thailand, no more details provided | 60–85 | Seroprevalence | Anti-HAV IgG antibody | Abbott | 93 | 98.92% |
| Poovorawan et al. [25] | Thailand | 2013 | 2012 | Outbreak Study | | Community-wide HAV outbreak | | Hospital based/ Buengkan province | 2–70 | Confirmed case | Anti-HAV IgM antibody | Abbott |
| 205* | 82.93% | | | | | | | | | | | |
| Linkins et al. [26] | Thailand | 2013 | 2006–2008 | Cross-Sectional | Men who have sex with men/ Single city | Men participating in the Bangkok Men who have sex with men Cohort Study | 18–45 | Seroprevalence | Anti-HAV total antibody | Murex Biotech | 1,291 | 27.03% |
| Sunthornchart et al. [27] | Thailand | 2008 | 2003–2005 | Cross-Sectional | Intravenous Drug Users/ Single city | Hospital based Bangkok | ≥20 | Seroprevalence | Anti-HAV total antibody | Diasorin | 1,107 | 60.16% |
| Samakoses et al. [28] | Thailand | 2007 | 2001 | Cross-Sectional | University students/ Single city | Students College of Medicine and the Nursing College of the Royal Thai Army | 17–26 | Seroprevalence | Anti-HAV total antibody | Abbott | 432 | 15.28% |
| Chatproedprai et al. [29] | Thailand | 2007 | 2004 | Cross-Sectional | Healthy Children, 6 months of age/ Nationwide | Hospital passed 4 provinces from north, north–east, center and south of Thailand | 0–85 | Seroprevalence | Anti-HAV total antibody | Abbott | 3,997 | 27.37% |
| Ratanasuwan et al. [30] | Thailand | 2004 | 2000–2002 | Cross-Sectional | Population based/ Nationwide | Six provinces in the Central Region of Thailand | all ages | Seroprevalence | Anti-HAV total antibody | Abbott | 1,514 | 52.25% |
| Jutavijittum et al. [31] | Thailand | 2002 | 1998–2000 | Cross-Sectional | Children, 4–16 years of age | Urban and rural schools School based/Single province | 4–16 | Seroprevalence | Anti-HAV total antibody | Sanofi Diagnostic Pasteur | 1,145 | 9.61% |

(Continued)

**Table 1.** (Continued)

| Author | Country | Publication year | Study year | Study design | Study population | Study setting | Age group (years) | Outcome | Clinical measurement | Platform manufacturer | Total number of samples tested | Seroprevalence |
|---|---|---|---|---|---|---|---|---|---|---|---|---|
| Pancharoen et al. [32] | Thailand | 2001 | 1998–1999 | Cross-Sectional | Population based/ Single city | Nurseries, school and graduate schools in Bangkok | 1–30 | Seroprevalence | Anti-HAV IgG antibody | Murex Biotech | 895 | 12.85% |
| Pramoolsinsap et al. [33] | Thailand | 1999 | 1998 | Cross-Sectional | Chronic Hepatitis B & C patients and healthy blood donors | Hospital based/ single city | 16–60+ | Seroprevalence | Anti-HAV IgG antibody | Abbott | 195 | Healthy donors 64.62% (126/195) Asymptomatic HBV carriers 68.6 (348/507) HBV-Related CLD (162/196) HCV-related CLD (106/117) |
| Pilakasiri et al. [34] | Thailand | 2009 | n.a. | Cross-Sectional | University students/ Single city | Nursing students at the Royal Thai Army Nursing College, Bangkok | 16–41 | Seroprevalence | Anti-HAV total antibody | Abbott | 381 | 8.92% |
| Luksamijarulkul et al. [35] | Thailand | 2003 | 1999–2000 | Cross-Sectional | Population based/ Single city | Residents hill-tribe communities in the north of Thailand. (rural area) | 15–24 | Seroprevalence | Anti-HAV total antibody | General Biologicals Corp | 190 | 87.89% |
| Chatchatee et al. [36] | Thailand | 2002 | 1996–2001 | Cross-Sectional | University students/ Single City | Students of medical college in Bangkok | 20–22 | Seroprevalence | Anti-HAV total antibody | Abbott | 135 | 11.11% |
| Poovorawan et al. [37] | Thailand | 2000 | n.a. | Cross-Sectional | Population based/ Nationwide | Participants of vaccination trial against hepatitis B virus infection collected in 5 representative provinces in Thailand | 1–18 | Seroprevalence | Anti-HAV Total antibody | Abbott | 961 | 7.91% |
| Rianthavorn et al. [38] | Thailand | 2011 | 2009–2010 | Cross-Sectional | Hospital based/ Single city | Tak province, border province between Thailand and Myanmar | all ages | Seroprevalence | Anti-HAV total antibody | Abbott | 308 | 70.78% |
| Hau et al. [39] | Vietnam | 1999 | 1999 | Cross-Sectional | Population based | Laboratory based community study | 0–87 | Seroprevalence | Anti-HAV IgG antibody | Abbott | 646 | 96.90% |
| Wahyuddin et al. [40] | Indonesia | 2019 | 2015–2016 | Case-Control | Healthy children | Regional, school based | 12–14 | Confirmed case | Anti-HAV IgM antibody | Chemux BioScience | 72 | 75.00% |
| Mohd et al. [41] | Malaysia | 2001 | 2000 | Outbreak Study | Population based | Residents of Kuala District, Terengannu state | 2–71 | Confirmed case | Anti-HAV IgM antibody | n.a. | 334 | 100.00% |
| Vengopalan et al. [42] | Malaysia | 2004 | 2002 | Outbreak Study | Population Based | Residents Hulu Langat District, Selangorstate | 1–40 | Confirmed case | Anti-HAV IgM antibody | n.a. | 51 | 100.00% |
| Yusoff et al. [43] | Malaysia | 2015 | 2012 | Outbreak Study | Population Based | Residents of Manjung District, Perak State | 13–72 | Confirmed case | Anti-HAV IgM antibody | n.a. | 78 | 100.00% |
| Poovorawan et al. [44] | Thailand | 2005 | 2002–2003 | Outbreak Study | Institution based/ Single city | Childcare center suburban area of Bangkok | 1–6 | Confirmed case | Anti-HAV IgG and anti-HAV IgM antibodies | Abbott | 112 | IgG: 66.07%; IgM: 62.50% |
| Phanwong et al. [45] | Thailand | 2008 | 2005 | Outbreak Study | Population based | Wiangpapao district, Chiang Rai Province | n.a. | Confirmed case | Anti-HAV IgM antibody | n.a. | 1,308 | n.a. |
| MOH, Thailand [46] | Thailand | 2019 | 2017 | Outbreak Study | Institution based/ Single city | Prisoners, Bangkok City | n.a. | Confirmed case | Anti-HAV IgM antibody | n.a. | 141 | 43.26% |
| MOH, Malaysia [47] | Malaysia | 2019 | n.a. | Surveillance system | n.a. | n.a. | All ages | Confirmed case | Incidence HAV rate | n.a. | n.a. | n.a. |
| DOH, Republic of Philippines [48] | Philippines | 2019 | n.a. | Surveillance system | Population based | Nationwide surveillance report | All ages | Confirmed case | Incidence HAV rate | n.a. | n.a. | n.a. |
| MOH, Singapore [49] | Singapore | 2018 | n.a. | Surveillance system | Population based | Nationwide surveillance report | All ages | Confirmed case | Incidence HAV rate | n.a. | n.a. | n.a. |
| MOH, Singapore [50] | Singapore | 2019 | n.a. | Surveillance system | Population based | Nationwide surveillance report | All ages | Confirmed case | Incidence HAV rate | n.a. | n.a. | n.a. |

(Continued)

**Table 1.** (Continued)

| Author | Country | Publication year | Study year | Study design | Study population | Study setting | Age group (years) | Outcome | Clinical measurement | Platform manufacturer | Total number of samples tested | Seroprevalence |
|---|---|---|---|---|---|---|---|---|---|---|---|---|
| MOH, Thailand [51] | Thailand | 2019 | n.a. | Surveillance system | Hospital based/ Nationwide | Nationwide surveillance report | All ages | Confirmed case | Incidence HAV rate | n.a. | n.a. | n.a. |

*Out of a total of 1,619 patients with clinical symptoms of hepatitis A who visited the hospital during the outbreak.

Abbreviations: CLD, chronic liver disease; DOH, Department of Health; HAV, hepatitis A virus; HBV, hepatitis B virus; IgG, immunoglobulin G; IgM, immunoglobulin M; MOH, Ministry of Health; n.a., not applicable.

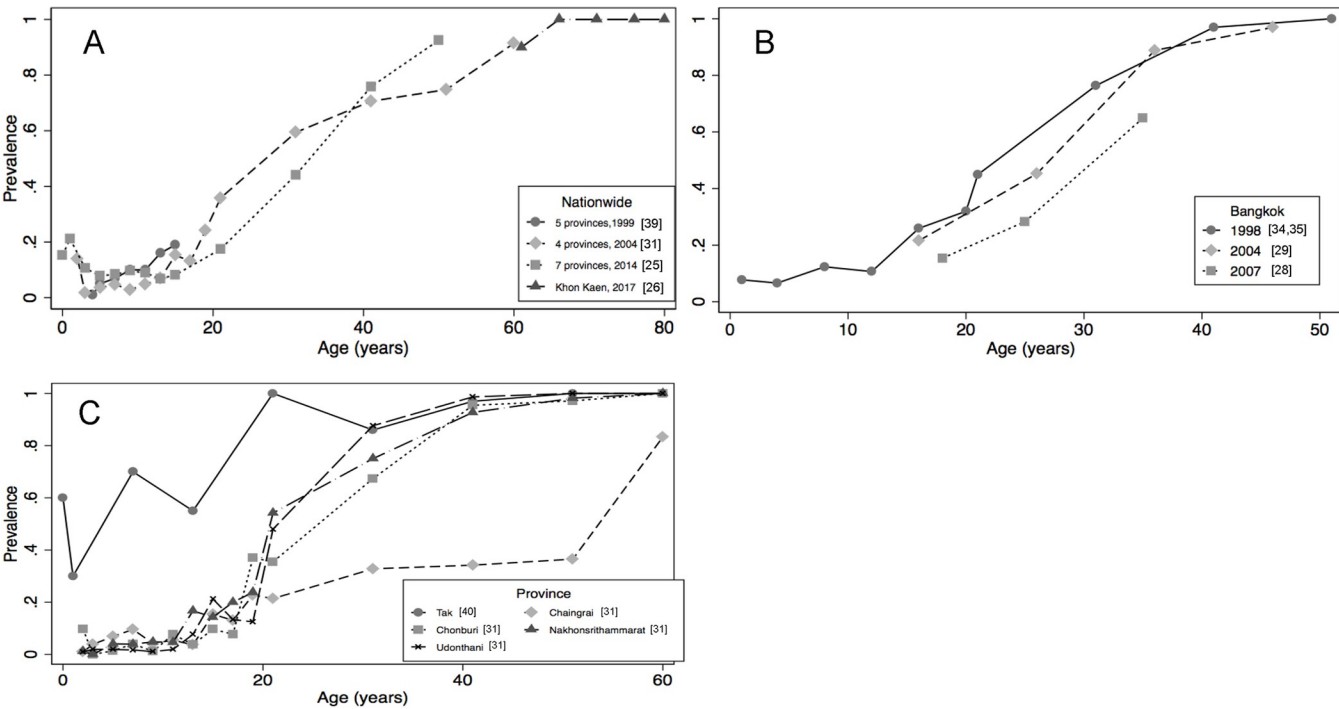

**Fig 2.** Age Specific HAV Seroprevalence[Y] in Thailand in Different Regions and Years A) Surveys with National Representativeness 1999–2014 B) Surveys from Bangkok 1998–2007 C) Surveys across Different Provinces 2004 (Except Tak Province Ran in 2010). [Y]Note: Seroprevalence measured by either detection of total HAV antibodies or specific IgG HAV in blood. Abbreviations: HAV, hepatitis A virus; IgG, immunoglobulin G.

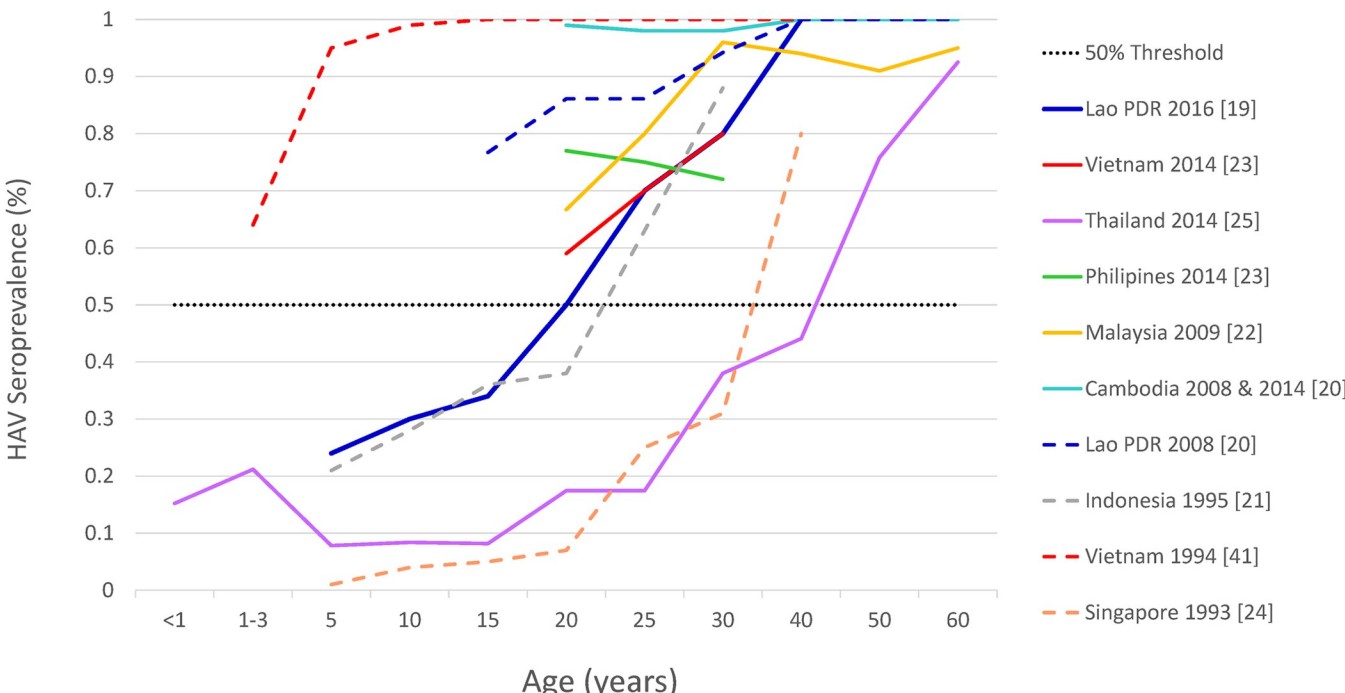

**Fig 3. Age specific trends in HAV Seroprevalence from ASEAN countries (1999–2020) (n = 10).** Note: The dotted line represents the 50% threshold to visually categorize HAV endemicity. Only latest information from Thailand is depicted intentionally. Abbreviations: ASEAN, Association of Southeast Asian nations; HAV, hepatitis A virus.

## Incidence of hepatitis A

Reports on the incidence of hepatitis A were found for four countries in Southeast Asia including Malaysia [47], Singapore [51, 52], Thailand [51] and the Philippines [48], where hepatitis A is a notifiable waterborne disease. Please refer to **S1 Fig** for a geographical overview of this review.

**Malaysia.** Hepatitis A has been a notifiable disease in Malaysia since the late 1980s [53] and since then there has been a sharp and sustained decreasing trend of the number of HAV cases, from around 9.0 per 100,000 individuals in 1991, to 2.2 in 2004, to the lowest level thus seen of 0.14 per 100,000 individuals in 2017. The latest report from 2018 showed a slightly increased incidence rate of 0.30 per 100,000 individuals [47].

**Singapore.** Official reports from the national health statistics, dating back to 1999 reported a stable range of notifiable cases (range: 48 [2016]– 146 [2006]) [49]. The latest report from 2018 and 2019 [50] recorded 75 and 66 new cases (crude annual incidence rates estimate approximately 1.0 per 100,000 population), respectively [50].

**Thailand.** Since 2003 National Health Statistics depicted stable incidence rates below 1.0 per 100,000 except in years 2005 and 2012 when incidence rates of 4 and 3 per 100,000 were estimated, respectively [51]. The latest report from 2019 stated the occurrence of 432 cases corresponding to an incidence rate of 0.65 per 100,000 [51].

**The Philippines.** Official reports from the national health statistics dating back to 2010 [48] reported a steady increase in annual cases from 440 to 563 in 2015. The two latest available reports from 2016 and 2017 showed 443 and 462 cases per year, respectively. Nearly 50% of the affected people were between 16–30 years of age. Estimated Crude annual incidence rates based on this information ranged between 0.4 and 0.6 cases per 100,000 population.

## Outbreak reports

Outbreak information was reported for Indonesia [40], Malaysia [20, 41–43], Singapore [22] and Thailand [23, 44, 45]. No information on outbreaks occurring in the Philippines was found in the search (**Table 1**). The outbreak in Indonesia (n = 59) occurred by the end of 2015, in two very close junior high schools in the urban area of Surabaya. In Malaysia at least 3 outbreaks from different regions were documented: The first happened in 2000 [41] where 334 cases of hepatitis A were reported and the outbreak source was postulated to be of mixed sources and spread from person to person. The second outbreak was reported in 2002 [42] with 51 cases of hepatitis A and the outbreak source was postulated to be contamination of rivers used for recreational purposes and human sewage. The last outbreak was reported in 2012 [43] with 78 cases, and the outbreak source was postulated to be contaminated "Toddy", an alcoholic beverage. In Singapore one outbreak attributed to consumption of raw shellfish resulted in 159 cases reported in 2002 [22]. The epidemiological report showed no specific food center to be implicated. In the other hand we found several reports from Thailand [23, 25, 44, 45]. The largest outbreak was reported in Buengkan province which affected more than 1,600 patients who attended the provincial hospital [25] and the latest outbreak report was from 2019, which reported 61 confirmed cases in a prison located in Bangkok [46].

## Risk of bias assessment

The result of the risk of bias assessment revealed that the studies included in the review showed high internal validity (**Fig 4**). Assessment scores ranked between low and medium risk according to the Hoy *et al.* scale [14], mainly affecting external validity. These studies showed flaws in sampling methodologies such as a lack of randomization or census procedures and applying convenience sampling strategies. Nine publications failed to provide confident representation

| Publication | 1.A close representation target population | 2.True or close representation entire population | 3.Random sample or census for sampling used | 4.Minimal non-response bias | 5.Directly from the subjects | 6.Acceptable case definition | 7.Reliability and validity | 8. The same mode of data collection | 9.The length of the shortest prevalence period | 10.Numerator and denominator properly used and given | Overall result |
|---|---|---|---|---|---|---|---|---|---|---|---|
| Posuwan et al, 2019 [26] | green | red | green | green | green | green | green | green | green | green | green |
| Kwon et al, 2018 [23] | red | red | red | green | green | green | green | green | green | green | green |
| Rianthavorn et al, 2011 [40] | green | red | green | green | green | red | green | green | green | red | amber |
| Sa-nguanmoo et al, 2016 [25] | green | red | green | green | green | green | green | green | green | green | green |
| Linkins et al, 2013 [28] | red | red | red | green | green | red | green | green | green | green | amber |
| Lee et al, 2011 [24] | green | green | green | green | green | red | green | green | green | green | green |
| Ahmad et al, 2011 [22] | red | red | red | green | green | red | green | green | red | green | amber |
| Chadproedprai et al, 2007 [31] | green | red | green | green | green | green | green | green | green | green | green |
| Chaiyaphruk et al, 2009 [36] | red | red | red | red | red | red | green | green | green | green | amber |
| Sunthornchart et al, 2008 [29] | green | red | red | green | green | red | green | green | green | green | amber |
| Samakoses et al, 2007 [30] | green | red | green | red | green | green | green | green | green | green | green |
| Ratanasuwan et al, 2004 [32] | green | red | green | green | green | green | green | green | green | green | green |
| Luksamijarulkul et al, 2003 [37] | red | green | red | green | green | green | green | green | green | red | amber |
| Jutavijjitum et al, 2002 [33] | red | red | green | green | green | green | green | green | green | green | green |
| Poovorawan et al, 2009 [20] | red | red | red | red | green | green | green | green | green | green | amber |
| Pancharoen et al, 2001 [34] | red | green | red | green | green | green | green | green | green | red | amber |
| Poovorawan et al, 2000 [39] | red | green | green | red | green | green | green | green | green | red | green |
| Juffrie et al, 2000 [21] | green | green | red | green | green | red | red | green | red | green | amber |
| Pramoolsinsap et al, 1999 [35] | green | red | green | green | green | green | green | green | green | green | green |
| Hau et al, 1999 [41] | green | red | green | green | green | green | green | green | green | green | green |
| Chatchatee et al, 2002 [38] | red | red | red | green | green | green | green | green | green | green | amber |
| Khounvisith et al, 2020 [19] | green | red | green | green | green | green | green | red | green | red | amber |

**Fig 4. Risk of bias assessment (n = 22¥).** ¥Risk of bias was not assessed for reports or publications from official Ministries of Health websites and outbreak specific studies. Color coding: Per item result: green = Yes; red = No. Overall result: green, and amber dots correspond to low and medium risk of bias, respectively.

of the targeted or the national population and six publications failed in the attempt to control the non-response bias.

## Discussion

This systematic review found evidences that suggest a full spectrum of HAV endemicity in Southeast Asia, where, aside from Thailand and the first recent study published from Lao PDR, none of the other countries have updated information on HAV seroprevalence at the national or regional level in the last 20 years.

Such lack of information might have failed to document the ongoing and expected HAV endemicity shift in these countries that have experienced major socioeconomic improvements in the past two decades [54].

The striking contrast in the number and size of HAV outbreaks in Thailand and Singapore, both countries with very low HAV endemicity profiles, illustrates the impact of a HAV endemicity transition with different pace over time. By 1993 and probably as a result of an earlier, faster, and greater socioeconomic development [55], Singapore already featured a very low HAV endemicity profile, while Thailand was just reaching intermediate levels (reaching the HAV exposure threshold of 50% among individuals 12 years of age) [23]. This example adds to the evidence that countries with an ongoing HAV endemicity shift, even those reaching low HAV endemicity levels, cannot rely solely on improvements in water hygiene and sanitation to contain the burden of hepatitis A infection [56, 57].

Although we only found one original outbreak report from Indonesia, there is evidence of several outbreaks across its territory. At least 47 episodes of hepatitis A outbreaks were documented between 1998 and 2018, mainly from the Isle of Java. Outbreaks were reported at schools and nearby areas (73%) and primarily involved people between 15–35 years of age [58]. Additionally, our review did not capture a recent outbreak report from July 2019 with over 950 cases in the Isle of Java [59] because it was in a press release format. These large numbers of outbreaks are underpinned by an ongoing transition of HAV endemicity that has not been documented yet. Lack of routine surveillance of hepatitis disease and low confidence in the existing epidemiological data in Indonesia has been previously reported [60]. The only HAV seroprevalence survey during our study period came from a single city (Yogyakarta) in 1996, showing a low HAV endemicity profile. As such, this data cannot be generalized to the entire Indonesian population [19].

This review did not find any published information on outbreaks in the Philippines, and very limited information on seroprevalence for the last two decades, although periodic surveillance reports of acute diseases have been published since 2010 showing incidence rates below 1 per 100000, similar to those reported from very low endemic countries in the region. The case of the Philippines shows that reporting up to date incidence data needs to be complemented with periodic seroprevalence surveys to fully capture the burden of HAV disease.

Among the 4 countries in Southeast Asia where hepatitis A is a notifiable disease, only Thailand provides updated information of HAV seroprevalence at a national level. This evidence confirms that there are clear gaps in the HAV epidemiology surveillance in the region. There is a clear need for establishing and improving nationwide HAV surveillance in most of the countries in Southeast Asia.

Seroprevalence studies captured from Malaysia [20] and Vietnam [21, 39] suggest that these countries might be at an intermediate endemicity level but fall short in determining the age at which a 50% threshold is reached (**Fig 3**). In other words, the existing evidence is limited in assessing the age at midpoint of population immunity, defined as the youngest age at which half of the population has serologic evidence of prior exposure to HAV.

HAV Seroprevalence age trends reports from Lao PDR, suggesting a low endemicity profile in both rural and urban areas, contrast with those observed in Cambodia and Myanmar despite sharing the lower Human Development Indices in the region [54]. This could be attributed to the improvements in water sanitation achieved by Lao PDR in the last few decades [61, 62].

The burden of hepatitis A among high-risk groups, merits further discussion. Information, mostly from Thailand, related to specific populations such as healthcare professionals, intravenous drug users and men who have sex with men shows results consistent with the general population [23]. These results emphasize the identical baseline susceptibility of this group to the general population, with a higher risk of acquiring HAV due to the inherent nature of their profession or behaviors [63]. Similar information was lacking from the other countries in the region.

HAV seroprevalence surveys including individuals from early childhood (around 5 years or even less, when possible) to late adulthood remains the most relevant data for HAV epidemiological analysis and disease modeling given the strong cohort effect shown by the HAV endemicity transition [64, 65]. It is likely that the young birth cohorts in Southeast Asian countries will remain at high risk of HAV infection, although with regional and temporal variations.

It is fully acknowledged that breaking HAV transmission and, in this case, ensuring improvement of water sanitation and hygiene (WASH) remains the cornerstone of HAV containment in a country, but to accomplish it country and region wide is not an easy task in Southeast Asia [66]. Thus, depending on the epidemiological pattern, countries may consider targeted vaccination against the disease to stop its spread, protect the individual with life-long immunity and induce herd immunity [67].

This systematic review addressed some of the limitations of previous publications reporting seroprevalence and incidence data from Southeast Asia. It reported a risk assessment bias and included both a literature search in the local language and grey literature search to rule out the existence of potential data being excluded from the mainstream search.

Scores of all seroprevalence papers included in this review ranked between low and medium risk, highlighting the reliability of the results included in the analysis. Most of the studies performed well in the internal validity items. The shortcomings mainly occurred over the external validity. This is not surprising in cross-sectional studies [68] as the generalization of a study outcome to a broader population is limited by the high variability across populations. Therefore, future studies should join efforts to reach local, regional, or national representatives to maximize the usefulness of HAV seroprevalence surveys.

Nevertheless, these results should be interpreted with caution owing to the limitations of the studies retrieved. Aside from Thailand, the low number of studies per country, their limited national representativeness and the heterogeneity of study population limit their external validity. In addition, the notable difference in the sampling methodology in the seroprevalence studies and differences in reporting and completeness of incidence data across countries in Southeast Asia hinder their comparability.

In summary (**Fig 5**), this review documents the contrasting HAV endemicity and outbreak profiles of the countries in the Southeast Asian Region. It highlights the fact that ASEAN countries, aside from Thailand, have a substantial gap of up-to-date and reliable information on HAV epidemiology. This lack of information makes it difficult to fully document the epidemiology shifts, if any, in HAV endemicity. It is expected that several of these countries transit to low HAV endemicity in the next decades, which could consequently be associated with a higher risk and burden of disease in older age groups. Robust evidence on the HAV incidence and seroprevalence status of the population through the strengthening of surveillance systems

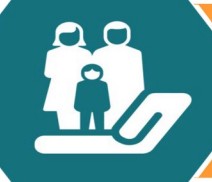

# Plain Language Summary

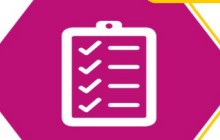

**What is the context?**

• Hepatitis A virus is transmitted through contaminated food and water or through close contact with an infected person. It usually occurs as an asymptomatic disease in children but may present with severe symptoms in adults causing temporary disability and absence from work.

• In 2009, the World Health Organization stated that there is limited evidence of the disease burden of hepatitis A from South East Asia, specifically, The Association of Southeast Asian Nations region which covers 11 countries (Brunei, Cambodia, Indonesia, Laos, Malaysia, Myanmar, the Philippines, Singapore, Thailand, Timor-Leste and Vietnam).

• We performed a systematic review of literature to collect and summarize information of HAV disease burden (new cases of HAV, incidence, and past exposure to HAV, and seroprevalence) over the last 20 years in South East Asia.

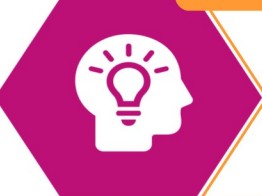

**What is new?**

We performed a systematic review of literature to collect and summarize information of hepatitis A disease burden (new cases of hepatitis A, incidence, and past exposure to hepatitis A, and seroprevalence) over the last 20 years in South East Asia. We show:

• a wide diversity of hepatitis A seroprevalence and incidence in the South East Asian region.

• the existence of sporadic outbreaks of different intensities in the region.

• most of the evidence found for South East Asia come from a single country: Thailand; information for the other countries is largely lacking or old.

• less than half the countries included in South East Asia have established hepatitis A surveillance.

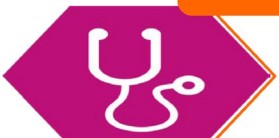

**What is the impact?**

• This review documents the need for further research to establish the current hepatitis A disease burden in the region.

• There is an urgent need to strengthen hepatitis A surveillance infrastructure and mechanisms in South East Asia.

• Up to date and reliable information on hepatitis A epidemiology will help to define specific strategies to control and reduce the burden of hepatitis A disease in South East Asia.

**Fig 5. Plain language summary.**

will be useful for decision-makers to define targeted and comprehensive strategies to reduce the current and future burden of HAV disease [69] and potentially help in achieving the ultimate goal of eliminating hepatitis A in Southeast Asia by 2030 [70].

## Supporting information

**S1 Checklist.**
(DOC)

**S1 Fig. Geographic representation of countries included in the review.**
(TIFF)

**S1 Text. Search strategy.**
(DOCX)

**S1 Table. Inclusion and exclusion criteria.** *References cited by screened articles were manually reviewed for relevance (i.e. snowballing). **References of included articles in these systematic reviews/meta-analyses were manually screened for additional relevant original articles (as deemed necessary by the reviewer).Abbreviations: HAV, hepatitis A virus; n.a., not applicable.
(DOCX)

## Acknowledgments

The authors would like to thank Business & Decision Life Sciences platform for editorial assistance and manuscript coordination, on behalf of GSK. Amandine Radziejwoski provided

editorial assistance and coordinated publication development. Amrita Ostawal (Arete Communication UG, Berlin, Germany, on behalf of GSK) provided medical writing assistance.

## Author Contributions

**Conceptualization:** Gustavo Hernandez-Suarez, Debasish Saha.

**Data curation:** Gustavo Hernandez-Suarez.

**Formal analysis:** Gustavo Hernandez-Suarez.

**Funding acquisition:** Gustavo Hernandez-Suarez.

**Investigation:** Gustavo Hernandez-Suarez, Debasish Saha, Kris Lodroño, Phatu Boonmahittisut, Stephanus Taniwijaya, Ashwini Saha, Selim Badur, Yong Poovorawan.

**Methodology:** Gustavo Hernandez-Suarez, Debasish Saha.

**Supervision:** Gustavo Hernandez-Suarez.

**Validation:** Gustavo Hernandez-Suarez.

**Visualization:** Gustavo Hernandez-Suarez.

**Writing – original draft:** Gustavo Hernandez-Suarez.

**Writing – review & editing:** Gustavo Hernandez-Suarez, Debasish Saha, Kris Lodroño, Phatu Boonmahittisut, Stephanus Taniwijaya, Ashwini Saha, Selim Badur, Yong Poovorawan.

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
