## [Decision Letter · Decision Letter 0]

31 May 2021

PONE-D-21-06795

Seroprevalence and incidence of hepatitis A in South East Asia: a systematic review

PLOS ONE

Dear Dr. Hernandez-Suarez,

Thank you for submitting your manuscript to PLOS ONE. After careful consideration, we feel that it has merit but does not fully meet PLOS ONE’s publication criteria as it currently stands. Therefore, we invite you to submit a revised version of the manuscript that addresses the points raised during the review process.

Your manuscript was reviewed by one expert in the field. Many reviewer have benn invited but could not accept invitation to review. The reviewer identified many important problems in your submission. Please review the attached comments and provide point-by-point responses.

We look forward to receiving your revised manuscript.

Kind regards,

Yury E Khudyakov, PhD

Academic Editor

PLOS ONE

Journal Requirements:

2. Thank you for providing the following Funding Statement: 

"GlaxoSmithKline Biologicals SA funded this review and all costs associated with its development and publication. GlaxoSmithKline Biologicals SA participated to the study design, data collection and analysis, decision to publish, and preparation of the manuscript. All authors had full access to all of the data in this study and take complete responsibility for the integrity of the data and accuracy of the data analysis.

The Center of Excellence in Clinical Virology, Chulalongkorn University supported Prof. YP."

We note that one or more of the authors is affiliated with the funding organization, indicating the funder may have had some role in the design, data collection, analysis or preparation of your manuscript for publication; in other words, the funder played an indirect role through the participation of the co-authors.

If the funding organization did not play a role in the study design, data collection and analysis, decision to publish, or preparation of the manuscript and only provided financial support in the form of authors' salaries and/or research materials, please review your statements relating to the author contributions, and ensure you have specifically and accurately indicated the role(s) that these authors had in your study in the Author Contributions section of the online submission form. Please make any necessary amendments directly within this section of the online submission form.  Please also update your Funding Statement to include the following statement: “The funder provided support in the form of salaries for authors [insert relevant initials], but did not have any additional role in the study design, data collection and analysis, decision to publish, or preparation of the manuscript. The specific roles of these authors are articulated in the ‘author contributions’ section.”

If the funding organization did have an additional role, please state and explain that role within your Funding Statement.

Please also provide an updated Competing Interests Statement declaring this commercial affiliation along with any other relevant declarations relating to employment, consultancy, patents, products in development, or marketed products, etc.  

3. We note that Figure 4 in your submission contain map images which may be copyrighted. All PLOS content is published under the Creative Commons Attribution License (CC BY 4.0), which means that the manuscript, images, and Supporting Information files will be freely available online, and any third party is permitted to access, download, copy, distribute, and use these materials in any way, even commercially, with proper attribution. For these reasons, we cannot publish previously copyrighted maps or satellite images created using proprietary data, such as Google software (Google Maps, Street View, and Earth). For more information, see our copyright guidelines: http://journals.plos.org/plosone/s/licenses-and-copyright.

3.1.    You may seek permission from the original copyright holder of Figure 4 to publish the content specifically under the CC BY 4.0 license. 

3.2.    If you are unable to obtain permission from the original copyright holder to publish these figures under the CC BY 4.0 license or if the copyright holder’s requirements are incompatible with the CC BY 4.0 license, please either i) remove the figure or ii) supply a replacement figure that complies with the CC BY 4.0 license. Please check copyright information on all replacement figures and update the figure caption with source information. If applicable, please specify in the figure caption text when a figure is similar but not identical to the original image and is therefore for illustrative purposes only.

Reviewers' comments:

Reviewer's Responses to Questions

**Comments to the Author**

1. Is the manuscript technically sound, and do the data support the conclusions?

Reviewer #1: Partly

2. Has the statistical analysis been performed appropriately and rigorously? 

Reviewer #1: No

3. Have the authors made all data underlying the findings in their manuscript fully available?

Reviewer #1: Yes

4. Is the manuscript presented in an intelligible fashion and written in standard English?

Reviewer #1: No

5. Review Comments to the Author

Reviewer #1: The authors conducted a conducted a systematic review to evaluate the existing literature regarding HAV epidemiology, focusing on its incidence and seroprevalence trends in the countries of the Southeast Asian region. This is an important study.

Major comments

1) Authors describe five database were used in Method section, however Only Medline is used in Flowchart. Please correct.

2) The authors descripted different publications period criteria between the Methods (January 1st, 1999 to February 15th, 2021) and the Table 1 (January 1999 to December 2019).

3) The objective of the study was focusing on the incidence and prevalence of HAV. The case-control study design was in included in the inclusion criteria. What was authors plan to do with the data from case-control study?

4) In Table 2, additional information on the characteristics of included studies is needed to better understand the results. Authors should consider including information such as study setting (rural, urban, nationwide?), studied population, reported or calculated seroprevalence. The discussion of the results should also consider these characteristics.

5) In Table 3, Reporting HAV endemicity solely based on the most recent survey may be misleading in classifying the country as low or high endemicity. The prevalence depends on the study subjects, setting and period of data collection.

6) The manuscript is unnecessarily long and does not allow for grasping the main points of the research. The authors should write concisely and precisely. Also, Figure 4 does not provide more information for this research. Authors should consider deleting it, or adding information related to the aim of the study.

Minor comments

1) In abstract, as authors have divided into different paragraph (Background, Methodology, Conclusion), it is better to mention results in an individual paragraph

2) In the methods: the authors should have screened the biography of all included publication as well. This procedure might help finding more relative publications.

3) The quality of figure 2 and figure 5 are too low to read.

4) In different paragraph of Study characteristics, they have put abbreviations in some cases in some not. It is better to put all the abbreviation together in beginning or end of the article.

5) “Prevalence of HAV” is very confusing. More suitable word, such as “HAV exposure” should be used.

6. PLOS authors have the option to publish the peer review history of their article (what does this mean?). If published, this will include your full peer review and any attached files.

Reviewer #1: No

---

## [Author Response · Author response to Decision Letter 0]

17 Aug 2021

Dr. Emily Chenette

Editor-in-Chief 

PLOS One 

Wavre, June 24, 2021

Subject: Revised submission of manuscript entitled: Seroprevalence and incidence of hepatitis A in Southeast Asia: a systematic review

Dear Dr. Chenette,

On behalf of my co-authors, I would like to resubmit to your attention the following manuscript for publication in PLOS One as a systematic literature review. This systematic review aims to collect information on hepatitis A virus (HAV) incidence and seroprevalence in select countries in the Southeast Asian region. I acknowledge that all comments received from the peer-review process have been adequately revised as requested by the reviewers. We have provided each of these comments, along with their respective revisions including line and page numbers alongside the revised manuscript.

I declare that all authors fulfil the criteria for authorship, have read and approved the current manuscript and agree with its submission to PLOS One. I confirm that this manuscript has not been published elsewhere and is not under consideration by any other journal.

On behalf of the authors, I look forward to hearing from you with your decision at your earliest convenience.

Kindest regards,

Gustavo Hernandez-Suarez

To the editors:

1. We would like to confirm that the manuscript meets PLOS ONE’s style requirements.

2. Both the financial disclosure and competing interests state and describe the funding organization and the authors’ affiliation with this organization. These statements can be found in the “additional information” file.

3. Figure 4 (now supplementary Figure 1) has been created for the purpose of this manuscript and does not contain images that may be copyrighted.

Reviewers' comments:

Reviewer's Responses to Questions

Comments to the Author

1. Is the manuscript technically sound, and do the data support the conclusions?

Reviewer #1: Partly

2. Has the statistical analysis been performed appropriately and rigorously?

Reviewer #1: No

3. Have the authors made all data underlying the findings in their manuscript fully available?

Reviewer #1: Yes

4. Is the manuscript presented in an intelligible fashion and written in standard English?

Reviewer #1: No

5. Review Comments to the Author

Reviewer #1: The authors conducted a conducted a systematic review to evaluate the existing literature regarding HAV epidemiology, focusing on its incidence and seroprevalence trends in the countries of the Southeast Asian region. This is an important study.

Major comments

1) Authors describe five database were used in Method section, however Only Medline is used in Flowchart. Please correct.

Response: Thank you for highlighting this point. Indeed, we conducted a search of five databases, MEDLINE (via PubMed), Embase, Google Scholar, the Health Research and Development Information Network (HERDIN) from The Philippines and MyJurnal from Malaysia. Among the studies included in this review, all studies were identified in Medline or through a grey literature search. We identified only duplicates among the studies found through searching other databases named above, we have edited the PRISMA flow chart (Fig 1) to describe the search results found on the remaining databases.

Line 172, page 10, Fig 1

2) The authors descripted different publications period criteria between the Methods (January 1st, 1999 to February 15th, 2021) and the Table 1 (January 1999 to December 2019).

Response: Thank you for your comment, the dates in table 1 (now supplementary Table 1) have been updated accordingly. This was incorrectly mentioned in table 1 as we had updated the search after 2019. 

3) The objective of the study was focusing on the incidence and prevalence of HAV. The case-control study design was in included in the inclusion criteria. What was authors plan to do with the data from case-control study?

Response: Thank you for your comment. We included case control studies in the eligibility criteria because of two reasons which are as follows: Case control study design is often use for outbreaks investigation and controls might be population based and can provide information on the frequency of past exposure of HAV in the general. One case control was included in this review to describe an outbreak in Indonesia.

4) In Table 2, additional information on the characteristics of included studies is needed to better understand the results. Authors should consider including information such as study setting (rural, urban, nationwide?), studied population, reported or calculated seroprevalence. The discussion of the results should also consider these characteristics.

Response: Thank you for your comment. We have edited Table 2 (now Table 1) to include further information on study setting, population and seroprevalence as recommended by the reviewer. These characteristics have also been described in the results section of the text. 

5) In Table 3, Reporting HAV endemicity solely based on the most recent survey may be misleading in classifying the country as low or high endemicity. The prevalence depends on the study subjects, setting and period of data collection.

Response: Thank you for your comment. We agree with the reviewer that the use of the most recent survey may not accurately describe the hepatitis A virus endemicity level of a country. As a result, we have decided to merge tables 2 and 3 together into one table (now Table 1) and removed the endemicity column from the new table entirely. Furthermore, we have edited the text in the discussion section to reflect the changes to the table 2 (now Table 1) by using words like ‘suggest’ rather than ‘show’ to interpret these results. 

6) The manuscript is unnecessarily long and does not allow for grasping the main points of the research. The authors should write concisely and precisely. Also, Figure 4 does not provide more information for this research. Authors should consider deleting it, or adding information related to the aim of the study.

Response: Thank you for your comment, we have edited the discussion section by removing certain text regarding a discussion of the risk of bias related to the included publications. This has been adequately described in the ‘risk of bias assessment’ sub-section in the results section. We also strived to reduce any other text throughout the manuscript which we deemed as repetitive or contrary to the point being discussed. Lastly, while PLOS One does not have any word limits for review submissions, the word count for this manuscript after revisions comes to approximately 4000 words excluding tables, figures and references, following the PRISMA guidelines that recommends 5000 word-limit for Systematic Reviews.

We acknowledge that the figure 4 (now supplementary Figure 1) was created from scratch by our team and that no copyright conflicts were identified upon its completion. The open source map of Southeast Asia was utilized to highlight the geographical context of the study. The authors found it important to include this figure as it would provide the unspecialized reader with an overview of the region. We have moved this figure to Supplementary Fig 1 and leave this matter to the discretion of the reviewers and journal editors on whether to include this figure or delete it from the manuscript. 

Minor comments

1) In abstract, as authors have divided into different paragraph (Background, Methodology, Conclusion), it is better to mention results in an individual paragraph

Response: Thank you for your suggestion. We have edited the abstract to include a paragraph which reports only the results of this review. This paragraph is now named ‘Results’ as recommended by the reviewer.

Line 39, page 3

2) In the methods: the authors should have screened the biography of all included publication as well. This procedure might help finding more relative publications.

Response: Thank you for your suggestion. The authors did screen the bibliographies of the publications included in this review, but it was not done systematically. This was because many countries had no other publications besides the ones already included in this review. Only Thailand had additional publications for which a bibliography screening was conducted. Manuscript text was edited in the methods and results sections, respectively.

3) The quality of figure 2 and figure 5 are too low to read.

Response: Thank you for highlighting this point. Upon checking the figures for quality standards described in the journal guidelines we can confirm that the figures adhered to the stipulated pixel limits. We have once again tried to provide the figures in a similar format.

4) In different paragraph of Study characteristics, they have put abbreviations in some cases in some not. It is better to put all the abbreviation together in beginning or end of the article.

Response: Thank you for your comment, we have revised the complete manuscript text in adherence to the journal guidelines. The guidelines stated that abbreviations must be mentioned in full at first use and only if used more than 3 times in the entire text. 

5) “Prevalence of HAV” is very confusing. More suitable word, such as “HAV exposure” should be used.

Response: Thank you for your comment. We have edited the manuscript text to ‘’HAV exposure’’ instead of ‘’seroprevalence’’ based on the context of the sentence.

---

## [Editor Report · Decision Letter 1]

4 Oct 2021

Seroprevalence and incidence of hepatitis A in Southeast Asia: a systematic review

PONE-D-21-06795R1

Dear Dr. Hernandez-Suarez,

We’re pleased to inform you that your manuscript has been judged scientifically suitable for publication and will be formally accepted for publication once it meets all outstanding technical requirements.

Kind regards,

Yury E Khudyakov, PhD

Academic Editor

PLOS ONE
---

## [Editor Report · Acceptance letter]

19 Nov 2021

PONE-D-21-06795R1 

Seroprevalence and incidence of hepatitis A in Southeast Asia: a systematic review 

Dear Dr. Hernandez-Suarez:

I'm pleased to inform you that your manuscript has been deemed suitable for publication in PLOS ONE. Congratulations! Your manuscript is now with our production department. 

Kind regards, 

on behalf of

Dr. Yury E Khudyakov 

Academic Editor

PLOS ONE